# Supervised Kohonen Self-Organizing Maps of Acute Asthma from Air Pollution Exposure

**DOI:** 10.3390/ijerph182111071

**Published:** 2021-10-21

**Authors:** Moses Mogakolodi Kebalepile, Loveness Nyaradzo Dzikiti, Kuku Voyi

**Affiliations:** 1Faculty of Health Sciences, School of Health Systems and Public Health, University of Pretoria, Pretoria 0083, South Africa; kuku.Voyi@up.ac.za; 2Faculty of Health Sciences, School of Health Systems and Public Health, Ross University School of Veterinary Medicine, Basseterre, Saint Kitts and Nevis; Loveness.dzikiti@gmail.com

**Keywords:** self-organizing maps, classification model, air quality, asthma outcomes, asthma research, artificial neural networks

## Abstract

There are unanswered questions with regards to acute respiratory outcomes, particularly asthma, due to environmental exposures. In contribution to asthma research, the current study explored a computational intelligence paradigm of artificial neural networks (ANNs) called self-organizing maps (SOM). To train the SOM, air quality data (nitrogen dioxide, sulphur dioxide and particulate matter), interpolated to geocoded addresses of asthmatics, were used with clinical data to classify asthma outcomes. Socio-demographic data such as age, gender and race were also used to perform the classification by the SOM. All pollutants and demographic traits appeared to be important for the correct classification of asthma outcomes. Age was more important: older patients were more likely to have asthma. The resultant SOM model had low quantization error. The study concluded that Kohonen self-organizing maps provide effective classification models to study asthma outcomes, particularly when using multidimensional data. SO_2_ was concluded to be an important pollutant that requires strict regulation, particularly where frail subpopulations such as the elderly may be at risk.

## 1. Introduction

South Africa is accepted as one of Africa’s most industrialized economies [1]. The rapid industrialization of South Africa has increased the demand for electrification and other inputs for industrial processes such as transportation and human capital [1]. The increased need for the latter has led to rural–urban migration and rapid urbanization. Consequent to rapid urbanization, industrialization and migration, social determinants of health, such as the place where people live, have also changed. As may be expected, new patterns of both communicable and non-communicable diseases have therefore emerged. One such disease with a significant economic and public health burden in South Africa is asthma. South Africa is ranked fifth in the world for age-adjusted mortality due to asthma [2]. Although not the only known trigger for asthma, pollution due to industrial processes and other anthropogenic processes has proved to be a concern for the South African context. In South Africa, as in many other emerging markets, the use of clean technologies to power economies is limited. Therefore, industrial processes generate increased levels of pollution, which in turn increase the incidence of acute asthma [2]. Nearly 85% of electricity generated in South Africa is generated through power plants using coal [1]. An additional 5% of the electricity is generated using diesel turbine engines [1]. These processes contribute to overall air pollution. Emissions from vehicles also contribute to pollution. Therefore, exposure to air pollution and the resultant acute exacerbations of asthma still form a pertinent research topic.

The relationship between environmental exposures and health has been the subject of public health research for decades [3,4,5]. There remain several unanswered questions regarding acute respiratory outcomes, particularly asthma, due to environmental exposures. Asthma is a heterogeneous disease with clinical presentation symptoms such as wheezing, shortness of breath, chest tightness and coughing, usually characterized by respiratory airway inflammation and hyper-responsiveness [6]. Research indicates that air pollution plays a role in exacerbation of asthma, but it remains unclear how air pollution causes asthma disease [7,8].

The role of aeroallergens in triggering asthma has been documented as early as the 1980s [9]. Although aeroallergens, particularly pollen, have been known to trigger asthma, studies in the South African context indicated that pollution was also associated with the increased incidence of acute asthma and hospitalization [2].

In West Africa, another study found that, in addition to industrialization, pollution generated by agricultural processes and burning of farm lands further contributes to exposure that increases respiratory symptoms and related mortalities [10]. Similar to the increasing incidence of asthma in relation to increased industrialization and pollution in sub-Saharan Africa, the same association has been documented for South East Asia [11]. The rapid industrialization of South East Asia has been linked to an increasing prevalence of asthma and allergies [11]. These findings linking increase in pollution to increased incidence of acute asthma underscore the importance of understanding the role of pollution in triggering non-allergic asthma.

Univariate analyses have produced knowledge on the association between exposure to pollution and asthma. However, alternative methods that enhance the understanding of exposures that precipitate acute exacerbations, to enable proactive and preventative interventions in the management of asthma, are required. The latter methods need to have independent assumptions generally required for traditional univariate statistical methods. These alternative methods need to be resilient to air quality data that are often sparsely available in emerging markets such as South Africa. Most importantly, proposed alternative methods need to be capable of describing non-linear relations between exposure to air pollution and the related health outcomes. The relationship between exposure and health may not necessarily be linear, as assessed in most univariate analysis methods, due to bio-variability and the complexity of mixed exposures. Non-linear methods such as ANNs have been found to perform better in forecasting compared to linear models such as the traditional Autoregressive Integrated Moving Average (ARIMA) model [12].

To assess this complex relationship between exposure to air pollution and health outcomes, the current study used a non-linear method, using artificial neural networks (ANNs). The study applied the Kohonen self-organizing neural network [13,14]. This is a dimension reduction algorithm that compresses the output of a multidimensional input into a two-dimensional grid or map called a self-organizing map (SOM). The algorithm in Kohonen’s self-organizing maps applies unsupervised learning; that is, it uses no assistance from external sources and, therefore, primarily performs clustering of training patterns or associations without known outcomes [15].

In the SOM model, the idea of self-organizing relates to the ability of systems to change their internal structure and function in response to external stimuli [15]. As the SOM produces a lower-dimensional projection of the high-dimensional input, the similarity relations between and within the input data points are preserved [15]. The SOM model closely resembles learning vector quantization (LVQ) and is based on competitive learning, where only the winning neuron learns [13].

The current study sought to contribute to understanding the attributes of environmental exposures and the characteristics of patients who presented with acute exacerbation of asthma. The primary research goal was to determine if acute exacerbations of asthma could be forecasted through a supervised machine learning algorithm to classify disease outcomes. The study hypothesized that a supervised neural network algorithm could be trained to cluster patients presenting with acute exacerbations of asthma, thereby allowing for extracting rules and attributes associated with mixed air pollution exposure that exacerbate asthma.

## 2. Materials and Methods

This was a cross-sectional study in two public hospitals serving the Johannesburg and Tshwane Metropolitan cities in Gauteng Province, South Africa. A map indicating the study area is submitted as supplemental submission A: Appendix A. The study used clinical records to collect information on patients who presented at the health facility with acute exacerbations of asthma and other respiratory ailments. The initial patient sample size was 1647 patients, but only 483 patients had complete address records. The degree and severity of asthma were not used as inclusion criteria, as the study only focused on incidence. The study excluded patients who had addresses that could not be geocoded. Only complete address records could be used to successfully geocode patients’ addresses for interpolation of air quality data received from the Department of Environmental Affairs (DoEA) air quality monitoring stations.

A self-developed data collection instrument with ethics approval (p460/2016) was used to collect clinical data and record environmental data. All the presenting clinical symptoms recorded on the medical records were also collected. This instrument is provided as Appendix A. In the analysis, all asthma cases were coded as 1, and non-asthma outcomes were coded as 0. Air quality data were requested from the DoEA for the City of Johannesburg (CoJ) and the City of Tshwane (CoT). Data from all the monitoring stations in the two cities were provided by the department. These data were pre-processed and transformed (standardization and scaling) before analysis. Data on nitrogen dioxide (NO_2_), sulfur dioxide (SO_2_), carbon monoxide (CO), ozone (O_3_), and particulate matter (PM_10_) were received. These data had varying degrees of missingness indicative of the sparsity of air quality data in South Africa.

Residential addresses in the clinical records were provided by patients without any form of verification. The patients’ addresses were geocoded in ArcGIS version 10.6. All demographic data were collected including gender, age, race and residential address. Age categories were defined as (a) category 1, infants (from birth to 1 year); (b) category 2, children (2–9 years); (c) category 3, adolescents (10–19 years); (d) category 4, adults (20–65 years); and (e) category 5, elderly (over 65 years of age). In the South African context, there are four race categories which were coded 0 to 3. Race category 0 was African or Black. Race category 1 was Whites. Race category 2 was Indian, and category 3 was Other.

Missing data were imputed using multiple imputation by chain equations (MICE). The completely imputed environmental data from the DoEA were then interpolated to the patients’ geocoded physical addresses.

As discussed in the introduction, an SOM method was preferred in the current study due to its ability to identify non-linear relations. It also offered better dimensional reduction and presentation of an output in two dimensions compared to other methods.

The first step in developing an SOM algorithm is deciding on the size of the map or grid. Selecting the right size of the map is important, as bigger maps with a high number of nodes are computationally heavy and may not always achieve sufficient data reduction [16]. In the current study, we used a size slightly bigger than the size recommended by the rule of thumb. We used an 8 × 8 grid in the final model because this size had an optimum model with lower quantization error. The SOM algorithm can have a rectangular, hexagonal or linear structure and uses the structure to organize the neurons [16]. The principal steps in developing an SOM included (a) initialization, (b) updating of codebook vectors, (c) updating codebook vectors using Euclidian distance and (d) computing the quantization error [15]. Initialization can be represented by Equation (1) below:*wkj* = (*wkj1*, *wkj2*, ···, *wKJI*)(1)

There are multiple ways to initialize the codebook vectors, but the simplest method is to assign random values to the weights as shown in Equation (1), where *K* represents the number of rows and *J* the number of columns.

Iterative updating of the codebook vector follows Equation (2) as
*wkj* (*t* + 1) = *wkj* (*t*) + *hmn*, *kj* (*t*)[*zp* − *wkj* (*t*)](2)
where *mn* is the row and column index of the winning neuron as determined by the Euclidean distance.

The iterative process of updating the codebook vectors ends when the minimum quantization error has been achieved. Equation (3) shows calculation of quantization error.
(3)$$\varepsilon t=\overset{pt}{\underset{p=1}{\sum }}\;||\;zp-wmn\left(t\right)\;||\;\left(\begin{array}{c} 2\\ 2 \end{array}\right)$$

To optimize the model performance, the learning rate is adjusted using Equation (4).
*η*(*t*) = *η*(*0*)*e*^− *t/τ2*^(4)

The SOM algorithm was best suited for the current study because it provided a clear classification map. Furthermore, the SOM algorithm could still work well using air quality data with high degrees of data missingness.

To optimize the SOM algorithm, we changed the learning rate. Equation (4) describes the computation for adjusting the learning rate. A learning rate is a parameter optimizer that ensures the gradient descent does not change too rapidly or too slowly. The smaller the rate, the slower the gradient descent, and the longer the iteration. If too big, it may also affect convergence of the model.

To perform SOM, the clinical data were merged with the interpolated DoEA pollution data. The exposure data used were exposures measured two days before a clinical presentation. This lag period was intended to allow for an inflammation process that often precipitated an acute exacerbation of asthma [17]. Lag periods of 0 to 5 days have been reported where patients presented with respiratory symptoms, such as asthma, after they were exposed to dust [18].

Further data pre-processing included scaling and normalization of the data to prevent the differences in the scales of the variables from affecting the output, producing a scale artifact (SA). Variables with a larger scale may appear to be more important or significant in the algorithm learning process compared to variables with a smaller scale. Therefore, scaling is essential and allows for the input values to fall within the active range of the activation function of the algorithm. Nominal input values were coded into binary input parameters in preparation for modelling.

In the current study, we performed both a supervised and an unsupervised SOM, with the former producing dependent variable codes indicative of class predictions, i.e., whether the candidate had asthma or not. The difference between outcomes of a supervised SOM and an unsupervised SOM is that the former can perform a regression (in the case where the known outcome variable is continuous) or a classification (in the case where the known outcome variable is binary or class type), while the latter purely performs clustering of the input data [19].

To improve convergence speed and SOM performance, we varied the learning rate and the optimization of neighborhoods. After performing the SOM, the trained weights did not have any boundaries indicating clusters. Two methods can be used to develop cluster boundaries on the trained weights: (a) using a unified distance matrix or (b) using a Ward clustering method [13]. We used the Ward clustering of the codebook vectors to visualize the cluster boundaries. Equation (5) shows how the Ward distance was calculated.
(5)$$drs=nrns\;nr+ns\;\left|\left|wr-ws\right|\right|\left(\begin{array}{c} 2\\ 2 \end{array}\right)$$

To evaluate the performance of the SOM algorithm, accuracy, quality, quantization and topological errors were computed. The quality of an SOM model is determined by the mean distance of objects mapped to a unit and consequently the mean distance of objects mapped to the codebook vector of that unit. The smaller the distances, the better the objects are represented by the codebook vectors. The distance between the neurons was mapped on the code book vectors, and the closer they are, the better. In this study, the effects of neighborhood size and the number of weights in the self-organizing map (SOM) on quantization error were also analyzed.

## 3. Results

Over 70% of the initial study population was lost to the study in final model development. These were patients whose addresses could not be geocoded and therefore did not have an exposure profile. Of the remaining sample (*n* = 483), about 53% (*n* = 255) were male, and nearly 2% (*n* = 7) had a missing gender identity in the medical records. Just under 20% (*n* = 92) of the clinical presentations were confirmed asthma cases, while others were either a case of respiratory tract infection (19%), cough and bronchitis (6%), or other related respiratory conditions.

The ages of the participants were rightly skewed, with the minimum age being four months and the eldest patient being 86 years old. Seventy-three percent of the patients were adults and the elderly. Over 53% of positive asthma cases were adults. The pronounced effect of age suggests frailty or higher susceptibility to acute disease outcome with advances in age. The effect of race in determining health outcomes was also evaluated.

Figure 1 and Figure 2 below show (a) the prevalence of asthma by age category and (b) the prevalence of asthma by gender. In Figure 1, the adulthood age category is shown to contribute to the highest count of participants who had asthma (10% of the total asthma cases). The elderly and adolescence age groups followed with a 4% contribution.

Although Figure 2 does not adjust for age category, it indicates that the positive asthma cases were mostly males (52%, *n* = 47).

The initial unsupervised SOM, as shown in Figure 3, demonstrated a successful reduction in the multidimensional inputs to the desired two-dimensional map on a 4 × 4 grid.

The output map is read from left to right and from the bottom to the top of the map. The bottom left cycle represents node 1, and the top right cycle represents the last node (node 16). In Figure 3, a 4 by 4 map, there are 16 nodes on which the various inputs are mapped. Except for node 1 and node 11, all the nodes showed that the different input parameters contributed to determining the disease class. Generally, gender, race and age were important in determining the disease class. The contributory factor of the input data is shown by the size of the fan, which is color specific for the input variable. In six of the nodes (nodes 2, 8, 9, 12, 14 and 16), environmental pollution parameters were deterministic of disease class, i.e., whether a patient had asthma or not.

As in the unsupervised SOM, the socio-demographic factors (age, gender and race) in the supervised SOM seemed to dominate most nodes. However, the effect of environmental pollution was better represented in the supervised SOM. The code plot of the supervised SOM in Figure 4 shows that the pollutants had equally contributing representations in many nodes. Nodes 1–6 (the bottom row of nodes) show that only the environmental pollutants determined the disease outcome for those patients. Nodes 8, 36, 40, 42, 43, 50, 51 and 53–64 show a balanced representation of the environmental parameters and the sociodemographic parameters. However, gender seemed to be the overall most predictive variable of disease outcome shown in nodes 15, 22–24, 29–32 and 37–40. SO_2_ demonstrated an equally predictive value as race. NO_2_ and PM_10_ seemed to have the least predictive value. However, PM_10_ was slightly more potent than NO_2_. In node 6, PM_10_ was the variable most predictive for classification of disease outcome, and NO_2_ was the most predictive in node 14.

The supervised SOM for classifying asthma disease outcome was successful. Appendix A show the dependent variable code plot. After optimizing the model performance indicators, the final model had an accuracy rate of 59%, a sensitivity rate of 18% and a specificity rate of 64%. Changing the learning rate in the SOM algorithm produced the most improvement in model performance. We tested the quality of the model using a quantization error. Distances less or equal to 0.05 were observed for 57 of the 64 nodes. Furthermore, a low quantization error was observed (0.03).

## 4. Discussion

The SOM output allowed the mapping of a complex interaction between gender, race and SO_2._ The model further showed the effect of NO_2_ and PM_10_. A traditional statistic model might have defined the odds ratios or the gradients that equate one parameter to the other in a linear function (points where known X values can be used to calculate the Y output values). However, the current SOM model established threshold values for individual parameters at every node. It is this type of outcome that makes unsupervised classification models such as SOM an alternative that may be useful in modelling complex relationships between exposure and disease outcome. If knowing outcomes (e.g., an odds ratio) was adequate in asthma research, it would be reasonable to posit that the burden of asthma in emerging markets such as South Africa would not result in the reported age-adjusted death rates. It is known what the odds of females presenting with more asthma may be, but additional information, such as the levels of SO_2,_ NO_2_ and PM_10_ that are sufficient to produce interactions that may precipitate acute asthma, remains the current challenge. SOM models can identify these interactions and quantify them.

An algorithm such as SOM that enables extracting learning rules even from sparse data, such as the type of missing air quality data observed in South Africa, has an advantage over other algorithms, particularly those that may be affected by missing and sparsely available data. Further development, i.e., algorithm soft computing, of the SOM-extracted rules allows for better prediction and prescription on the management of disease outcomes. The latter attribute made an SOM a strongly suitable model to predict disease outcomes given the clinical presentations of patients with respiratory conditions that might have been acute exacerbations of asthma or a related outcome. Based on the SOM developed in the current study, a soft computed algorithm was prototyped into an asthma device. A patent on the device is currently under review.

The prevalence of asthma in South Africa has been recorded to be on the rise [2]. The results of the International Study of Asthma and Allergies in Childhood (ISAAC) in two South African cities (Cape Town and Polokwane) have shown the prevalence to range between 16% and 20% for the 13 to 14-year-old age group [2]. Age in the current study was reported as skewed. Therefore, an age-adjusted prevalence was not suitable to report. However, an overall incidence of asthma was reported as 19% (*n* = 483).

More males had asthma compared to females (just under 5% more males had asthma). This was inconsistent with expectation. Previous studies have shown asthma to be more prevalent in females, although this difference may be age related [20]. Although the current study did not measure the incidence of occupational asthma, this type of asthma may also be useful to explain the finding that the current study had more males with asthma than females. We observed older patients to be more susceptible to acute asthma compared to other age groups. This finding is consistent with the literature. Old age has been shown to increase mortality in patients aged above 55 years old, although some studies indicate childhood asthma to be often severe [21].

We observed an interaction between environmental factors and sociodemographic factors in some nodes. This interaction suggests that knowing environmental attributes and managing environmental pollution is vital for asthma management in the South African context. In the supervised SOM, considering pollutants only, SO_2_ seemed to be a potent predictor for positive asthma cases. PM_10_ and NO_2_, although predictive, had a lesser contribution. This observation concurs with Greenberg et al., who observed that the probability of mild and moderate-to-severe asthma was greater in areas with high exposure levels of SO_2_ compared to sites with high levels of NO_2_ [22]. In that study, the probable reasons for different presentations of asthma severity with the exposure to the two pollutants are given in depth, and the reasons relate to the pathophysiology in both the lower respiratory space and the upper airways [22].

In this study, we modeled pollution variables that may be generated by industrial processes, and we found SO_2_ to be the most important. As it has been reported that 85% of electricity generated in South Africa is produced using coal as the fuel [1], the finding that SO_2_ had more potency in classifying asthma suggests a policy need to further regulate emissions such as SO_2_. Furthermore, compliance with new WHO standards becomes an important national imperative for South Africa. Rapid industrialization, urbanization and the migration of large populations of people to urban centers may lead to anthropogenic activities that increase generation of pollutants such as SO_2_.

## 5. Conclusions

Gender, race and age showed an association to clinical presentation with acute exacerbations of asthma. SO_2_ predicted acute asthma better than both NO_2_ and PM_10_. Although the initial study sample size was reduced due to unavailable exposure profiles, the self-organizing map still showed satisfactory performance. In this study we observed a low quantization error. This study concluded that SOMs present an opportunity for studying non-linear health outcomes. This is particularly true in environmental and public health where the burden of chronic diseases such as asthma is on the rise. This conclusion was strengthened by the ability of the model to successfully classify asthma outcomes, even when the geocoding process had resulted in the loss of a sizable proportion of the initial sample size. Multidisciplinary approaches used in this study showed resilience to missing data. Therefore, using an SOM is recommended in exposure assessment, particularly when clinical and environmental data may be sparsely available. As indicated, the air pollution data had varying degrees of missing data and were imputed. The SOM was still possible using data sets with missingness.

## 6. Limitation

The current study had limitations due to patients’ addresses that could not be geocoded. The addresses had no available georeferenced data and could not be processed. Therefore, the addresses that could not be geocoded could not be included in the study because, without geocoding, the exposure from the monitoring stations could not be interpolated to the patients’ addresses. The impact of this limitation is a loss of 70% of the study sample size. This reduction in sample size may have affected the power of the study, although sample size may not be a critical rate-limiting step in machine learning model development.

## Figures and Tables

**Figure 1 ijerph-18-11071-f001:**
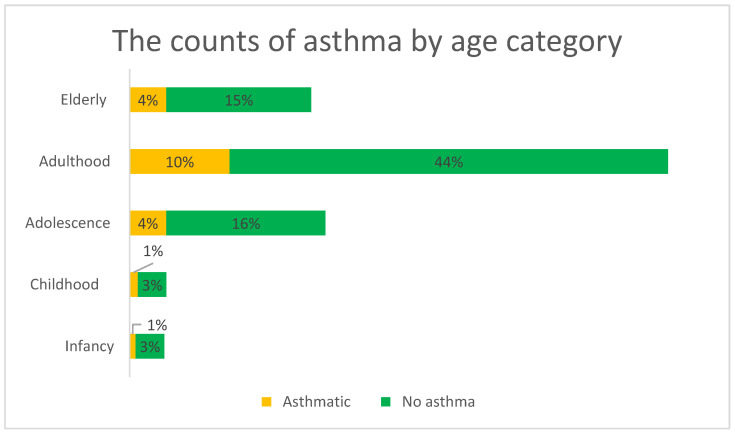
The counts of asthma by age category.

**Figure 2 ijerph-18-11071-f002:**
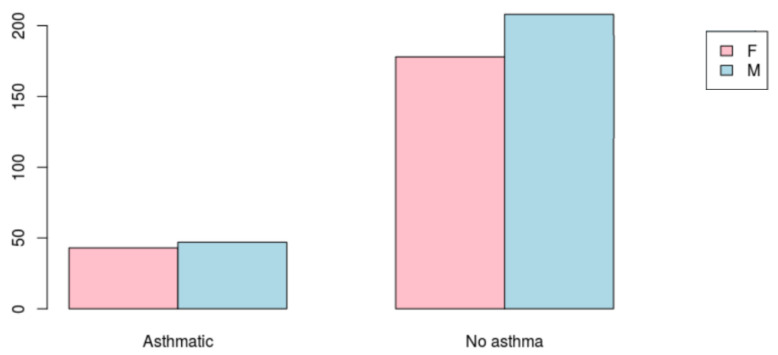
The counts of asthma by gender.

**Figure 3 ijerph-18-11071-f003:**
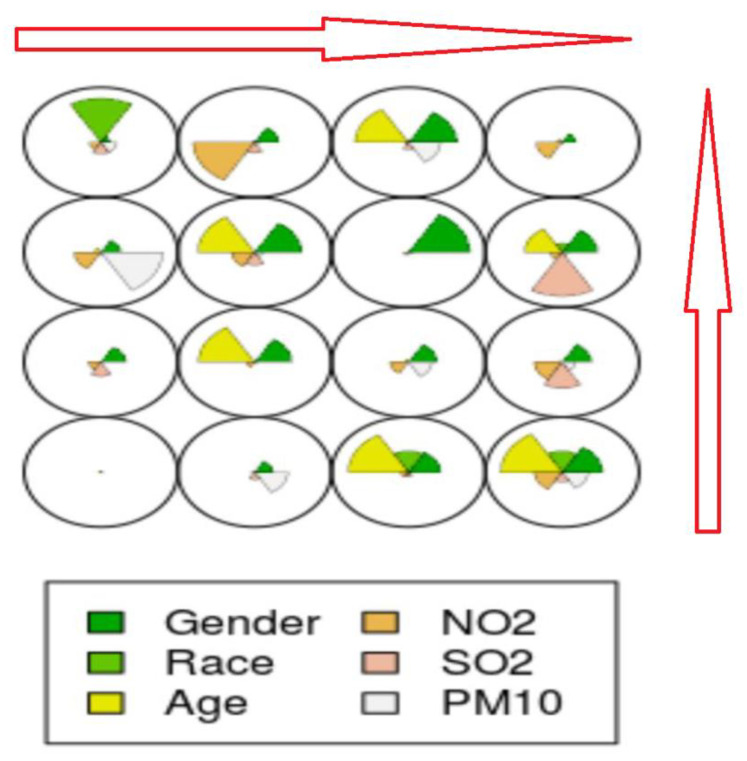
The code plot of an unsupervised SOM showing a two-dimensional map of the parameters determining disease outcome.

**Figure 4 ijerph-18-11071-f004:**
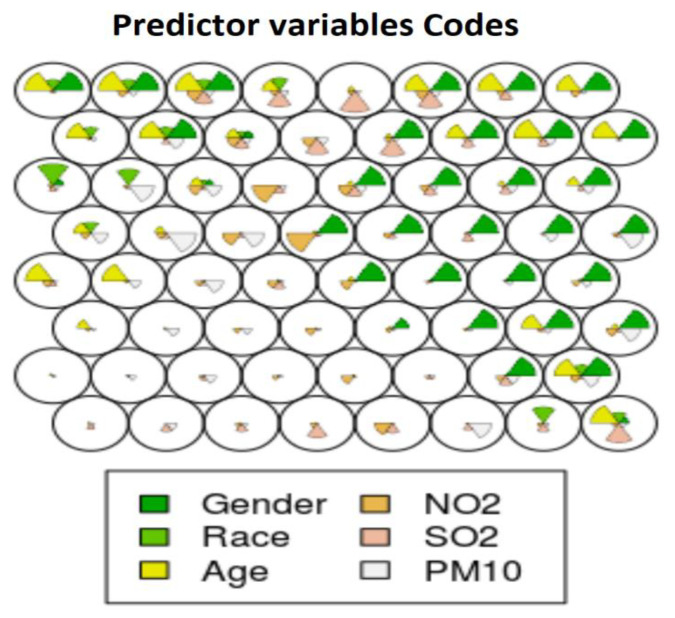
Code plot of the supervised SOM showing the independent variables.

## Data Availability

The data and all codes (R software codes and Python software codes) are readily available on request to the authors.

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
