# Peer review of "Supervised Kohonen Self-Organizing Maps of Acute Asthma from Air Pollution Exposure"

_ijerph, 2021, doi:10.3390/ijerph182111071_

Round 1

Reviewer 1 Report

I read the manuscript, 'Supervised Kohonen self-organizing maps of acute asthma from air pollution exposure' with great interest. The study sounds interesting, however, I think the main purpose of this complex study is not properly highlighted. I have some comments for the authors as the followings:

  1. The abstract has an unnecessary large introduction section and little about the methods, results, and conclusion. It is important to provide a reasonably good amount of information to be put for each of these sections. I strongly recommend rewriting the abstract with more outcome-oriented information. The hypothesis is also a must and has to be clearly mentioned.
  2. The 'Introduction' section needs to be revised properly highlighting what is already known and what is missing in the previous literature.  
  3. I am not sure about the section "2. Background on Asthma, air pollution and exposure assessment methods" in the introduction. This section is not important as it is only just a broad overview of the literature and is not pertinent to the research question. I suggest removing the entire section.
  4. The 'Methods' section is very short-fragmented without adequate information. For example, 3.1 to 3.4 could be clubbed into a single paragraph with a bit more details such as the names or number of cities from where the data were collected, severity fo the asthma patients (whether there were different asthma severities or all of them belonged to once category, etc.)
  5. The data analysis section starts with SOM. The question is why it was important to perform SOM. What about other univariate analyses? I would recommend a restructuring of the methods section.
  6. Line 341-342: It is important to describe the actual numbers in the results rather than just citing the figures. Also, what about the comparison? That was not mentioned.
  7. Figure 3: It is difficult to understand how to read the plot. Although the authors described it, it is quite difficult to understand. i would recommend adding an arrow horizontally at the top and vertically in the side to show the reading pattern. Otherwise, it is hard to find any of the numbered bubbles.
  8. Figure 4: I am not sure whether the figure became smaller and pixelated due to the formatting, but this is impossible to read.
  9. Figure 6: I am not sure what the plot adds to the manuscript. I find it hard to understand the importance of providing the clusters just for the dependent and independent variables. So as Figure 7, and I could not find this plot relevant to the research question addressed.
  10. Discussion section is quite a setback. I could not find the justification of the created models in this section. Again, it is important to compare previous observations with the current study which was missing.

Author Response

  1. The abstract has an unnecessary large introduction section and little about the methods, results, and conclusion. It is important to provide a reasonably good amount of information to be put for each of these sections. I strongly recommend rewriting the abstract with more outcome-oriented information. The hypothesis is also a must and has to be clearly mentioned.

Abstract was revised to include the recommendations.

  1. The 'Introduction' section needs to be revised properly highlighting what is already known and what is missing in the previous literature.  

Introduction revised as recommended.

  1. I am not sure about the section "2. Background on Asthma, air pollution and exposure assessment methods" in the introduction. This section is not important as it is only just a broad overview of the literature and is not pertinent to the research question. I suggest removing the entire section.

The section was removed. Only the portion introducing the machine learning algorithms used I the study was left.

  1. The 'Methods' section is very short-fragmented without adequate information. For example, 3.1 to 3.4 could be clubbed into a single paragraph with a bit more details such as the names or number of cities from where the data were collected, severity fo the asthma patients (whether there were different asthma severities or all of them belonged to once category, etc.)

The requested changes were made.

  1. The data analysis section starts with SOM. The question is why it was important to perform SOM. What about other univariate analyses? I would recommend a restructuring of the methods section.

The question was addressed in the introduction, and the discussions sections.

  1. Line 341-342: It is important to describe the actual numbers in the results rather than just citing the figures. Also, what about the comparison? That was not mentioned.

Comment addressed by including the recommended information.

  1. Figure 3: It is difficult to understand how to read the plot. Although the authors described it, it is quite difficult to understand. i would recommend adding an arrow horizontally at the top and vertically in the side to show the reading pattern. Otherwise, it is hard to find any of the numbered bubbles.
  2.  

The reviewer’s recommended improvement was applied.

  1. Figure 4: I am not sure whether the figure became smaller and pixelated due to the formatting, but this is impossible to read.

The figure was removed.

  1. Figure 6: I am not sure what the plot adds to the manuscript. I find it hard to understand the importance of providing the clusters just for the dependent and independent variables. So as Figure 7, and I could not find this plot relevant to the research question addressed.

The queried figures were removed.

  1. Discussion section is quite a setback. I could not find the justification of the created models in this section. Again, it is important to compare previous observations with the current study which was missing.

Comment addressed by revising the discussion section.

Reviewer 2 Report

This paper discussed the relationship between air pollution exposure and acute asthma. It deals with an important and interesting topic, namely supervised kohonen self-organizing maps of acute asthma from air pollution exposure.

Here are the suggestions for revision.

  • L43, L154, this sentence “it is difficult to link…… ” is duplicated.
  • L249 -251: The location map of study area in this paper should be supplement.
  • L215: “the first step in developing an SOM algorithm is deciding on the size of the map,” which size of this study are adopted? Why choose this size to study?
  • L350,368 with Figure 3 and 4, it is difficult to understand the situation in South African context.
  • L408, 434 with discussion and conclusion should be combined together, if can provide the policy implications of self-organizing maps of acute asthma will be better.
  • 1 and Fig.2 are too large, it would be better if carefully checked.
  • Please check references carefully.
  • The manuscript needs carefully check. Such as L290, 291 appears,43,44, L316 appears 37, and so on.

Author Response

Here are the suggestions for revision.

  • L43, L154, this sentence “it is difficult to link…… ” is duplicated.
  •  

The duplication was removed.

  • L249 -251: The location map of study area in this paper should be supplement.

The location map was developed and is submitted as supplement.

  • L215: “the first step in developing an SOM algorithm is deciding on the size of the map,” which size of this study are adopted? Why choose this size to study?

The study grid size and rational was included in the manuscript.

  • L350,368 with Figure 3 and 4, it is difficult to understand the situation in South African context.
  •  

The figures are on the South African data and therefore reflect the South African context.

  • L408, 434 with discussion and conclusion should be combined together, if can provide the policy implications of self-organizing maps of acute asthma will be better.
  •  

Policy implications discussed

  • 1 and Fig.2 are too large, it would be better if carefully checked.
  •  

Figures resized.

  • Please check references carefully.

References re-checked.

  • The manuscript needs carefully check. Such as L290, 291 appears,43,44, L316 appears 37, and so on.
  •  

Checked as recommended.

Reviewer 3 Report

The authors must be take in consideration not only the chemical aggressives, but also the vegetable ones in the spread of allergies and asthma., such as the parietaria. I recommend to read this paper: Palagiano C., Cenci G.P:, Pezzella M., Arena G., Parlato D., 1984, Analysis of the Respiratory Allergopathies of the Children in Rome,, pp. 245-254, in "Geographie et Santé. Actes du Symposium de Géographie de la Santé, 25th Congrès Intern. de Géographie, Paris-Alps 1984, Montpellier 21-26 August 1984, pp. 699.

Author Response

The authors must be take in consideration not only the chemical aggressives, but also the vegetable ones in the spread of allergies and asthma., such as the parietaria. I recommend to read this paper: Palagiano C., Cenci G.P:, Pezzella M., Arena G., Parlato D., 1984, Analysis of the Respiratory Allergopathies of the Children in Rome,, pp. 245-254, in "Geographie et Santé. Actes du Symposium de Géographie de la Santé, 25th Congrès Intern. de Géographie, Paris-Alps 1984, Montpellier 21-26 August 1984, pp. 699.

The reference was not readily accessible/found. It is identified as a conference proceeding, and it was found as a summary of the presentation, without the actual presentation. This summary was used in the revised manuscript. Furthermore, an additional study with a south African context was also utilized.

Round 2

Reviewer 1 Report

I thank the authors for amending my suggestions. I am happy with the revision.

Author Response

We thank the reviewer for the invaluable input that has improved the quality of our paper. The review was fair, informative, and greatly constructive.